# Repeated species radiations in the recent evolution of the key marine phytoplankton lineage *Gephyrocapsa*

El Mahdi Bendif [1], Bruno Nevado [1], Edgar L.Y. Wong [1], Kyoko Hagino[2], Ian Probert[3], Jeremy R. Young [4], Rosalind E.M. Rickaby[5] & Dmitry A. Filatov [1]

Phytoplankton account for nearly half of global primary productivity and strongly affect the global carbon cycle, yet little is known about the forces that drive the evolution of these keystone microscopic organisms. Here we combine morphometric data from the fossil record of the ubiquitous coccolithophore genus *Gephyrocapsa* with genomic analyses of extant species to assess the genetic processes underlying Pleistocene palaeontological patterns. We demonstrate that all modern diversity in *Gephyrocapsa* (including *Emiliania huxleyi*) originated in a rapid species radiation during the last 0.6 Ma, coincident with the latest of the three pulses of *Gephyrocapsa* diversification and extinction documented in the fossil record. Our evolutionary genetic analyses indicate that new species in this genus have formed in sympatry or parapatry, with occasional hybridisation between species. This sheds light on the mode of speciation during evolutionary radiation of marine phytoplankton and provides a model of how new plankton species form.

[1] Department of Plant Sciences, University of Oxford, Oxford OX1 3RB, UK. [2] Center for Advanced Marine Core Research, Kochi University, Nankoku, Kochi 783-8502, Japan. [3] Sorbonne Université - CNRS, Roscoff Culture Collection, FR2424 Station Biologique de Roscoff, 29680 Roscoff, France. [4] Department of Earth Sciences, University College London, London WC1E 6BS, UK. [5] Department of Earth Sciences, University of Oxford, Oxford OX1 3AN, UK. Correspondence and requests for materials should be addressed to D.A.F. (email: Dmitry.Filatov@plants.ox.ac.uk)

Marine phytoplankton play an important ecological role underpinning food webs in the ocean and are responsible for about half of global primary productivity[1]. Surprisingly, however, little is known about how marine plankton species originate and evolve[2–4]. Adaptation and speciation processes may work in rather different ways in relatively small subdivided populations of terrestrial organisms and globally ubiquitous populations of abundant marine plankton. In particular, it is unclear how new plankton species form in relatively homogenous habitats, such as in the open ocean[2], where no physical barriers to gene flow exist to promote allopatric speciation (i.e. the passive divergence of isolated populations) that is considered to be the most common speciation scenario in terrestrial organisms[5,6]. Ecological reasons for the 'paradox of the plankton'[7], the unexpected diversity of plankton species in a seemingly homogenous environment, have been analysed in multiple studies[8,9], but still remain obscure in terms of evolutionary genetic processes.

The calcifying coccolithophores (Haptophyta) represent an excellent model group to study the evolutionary processes underlying plankton speciation by integrating genomic, biological, biogeographic and palaeontological data. This group comprises around 200 well-described extant species which are widely distributed in modern oceans[10]. Coccolithophores form calcium carbonate scales, called coccoliths, which have complex and distinct architectures, allowing identification of morphospecies in extant diversity and the fossil record. Since their appearance around 220 Ma, coccoliths have formed massive calcareous deposits (e.g. the White Cliffs of Dover in southeast England) that serve as a sedimentary buffer of ocean chemistry and a major long-term carbon store that has significantly affected the global carbon cycle and Earth climate[11,12]. Marine pelagic sediments provide an enormously abundant and essentially continuous fossil record of coccoliths, and this has been extensively studied by biostratigraphers[13]. Based on coccolith morphology, the emergence and extinction of many coccolithophore morphospecies in the fossil record appear to be globally synchronous[14], though asynchronous events have also been recorded[15]. The causes of globally synchronous species appearances and disappearances are not well known[16].

In order to analyse the evolutionary processes underpinning macroevolutionary patterns in the plankton fossil record, we focus here on the coccolithophore genus *Gephyrocapsa* Kamptner (Noëlaerhabdaceae) that has a very abundant Pleistocene fossil record and shows exceptionally rapid species turnover[17]. Based on the fossil record, *Gephyrocapsa* shows a pattern of repeated coccolith size enlargement and abrupt coccolith size reduction through the Quaternary, which is consistent with similar size changes in their relatives *Reticulofenestra* throughout the Cenozoic[18–20]. The gradual size increase may be an instance of the Cope's rule, i.e. the trend of organism size increase within a lineage with evolutionary time, which was first described for terrestrial animals[21], but has also been reported in the marine realm[22]. Similar patterns of size shift over time have been reported in other marine protist groups, including diatoms[23], dinoflagellates[24] and planktic foraminifera[25]. The pattern is particularly well-marked in the genus *Gephyrocapsa*[17], though the evolutionary processes underpinning the pulses in coccolith size change remain unresolved.

Here we combine stratophenetic datasets on the *Gephyrocapsa* fossil record[17] with genome sequence data from a diverse set of 10 strains isolated across the world oceans to integrate evolutionary analysis of detailed fossil records and extant patterns of genetic variation. Based on whole-genome sequences, we reconstruct the phylogeny for five closely-related Noëlaerhabdaceae species and infer their evolutionary history and likely mode of speciation. The combined analysis of fossil record and genome sequence data reveals evidence for repeated species radiations and extinctions in this lineage through the Quaternary.

## Results and discussion

**Genome sequencing resolves *Gephyrocapsa* phylogeny.** Our analysis included the type species of the *Gephyrocapsa* genus, *G. oceanica*, that is widespread (Fig. 1a) and relatively straightforward to isolate and culture, as well as three other *Gephyrocapsa* species (*G. muellerae*, *G. parvula* and *G. ericsonii*; Fig. 1b), that were only recently isolated via a high-throughput sorting approach[19,20]. The analysis also included the most widely distributed and abundant coccolithophore species in the modern oceans, *Emiliania huxleyi* (Fig. 1b), that is very closely related to the *Gephyrocapsa* genus[26–28] and, as we show below, should be included in this genus. Previous attempts to resolve the phylogenetic relationships between members of *Gephyrocapsa* and *E. huxleyi* resulted in rather ambiguous patterns demonstrating conspicuous conflicts between molecular markers and interpretations based on morphological and palaeontological evidence[20,27–29]. In order to resolve phylogenetic relationships and analyse evolutionary processes driving speciation in this ecologically important phytoplankton genus, we sequenced the whole genomes of 8 *Gephyrocapsa* strains isolated from across the world oceans and combined these data with 2 other publicly available genomes (Supplementary Table 1). Based on morphology and previous studies[19,20], these 10 *Gephyrocapsa* strains were classified as belonging to five different species (Fig. 1b and Supplementary Fig. 1), covering almost the full range of extant morphological diversity in this genus[20].

Sequence alignments based on whole-genome sequences were used to reconstruct fully resolved phylogenies for five *Gephyrocapsa* species. Regardless of the phylogenetic reconstruction method used, topologies of resulting phylogenies were identical and all nodes in the trees were highly supported (100% bootstrap; Fig. 2a and Supplementary Figs. 2-4). The consensus phylogeny reconstructed from genome sequence data closely matches morphological taxonomy, with nearly all strains of the same morphospecies clustering together. This indicates that morphology-based definitions of *Gephyrocapsa* species (as in Young et al.[10]) correspond to biological species. This is consistent with a previous analysis of multiple *E. huxleyi* strains sampled across the world oceans that demonstrated genetic cohesiveness of this species[30]. The only mismatch of the genome-based phylogeny with morphology-based definition of species was the clustering of *G. parvula* strain RCC4033 with *G. ericsonii* strains RCC4032 and RCC4033 in the shallow *parvula/ericsonii* clade. However, even this result is supported by morphological evidence[31,32] since these two species, in fact, represent a complex of intergrading morphotypes that frequently co-occur in the water column and in the sediments of eutrophic tropical oceans. Based on the very small genetic distance between *G. parvula* and *G. ericsonii*, compared to other distances in the phylogeny (Fig. 2a), we suggest that these two morphospecies are likely to be conspecific. Consistent with the previous studies[19,20,26,27], our phylogenomic results support the use of the species name *Gephyrocapsa huxleyi* instead of the commonly used *Emiliania huxleyi* as this lineage is nested within the *Gephyrocapsa* genus (Fig. 2a). Hence, we hereafter use *G. huxleyi* instead of *E. huxleyi*.

Despite the clear clustering of the *Gephyrocapsa* strains by morphospecies and the 100% bootstrap support for all nodes in the tree (Fig. 2a), a considerable proportion (11.6%) of individual gene trees reconstructed from different 10 kb parts of the genome showed incongruence with the species tree in the densiTree plot (Fig. 2b). In particular, Shimodaira–Hasegawa tests[33] showed that

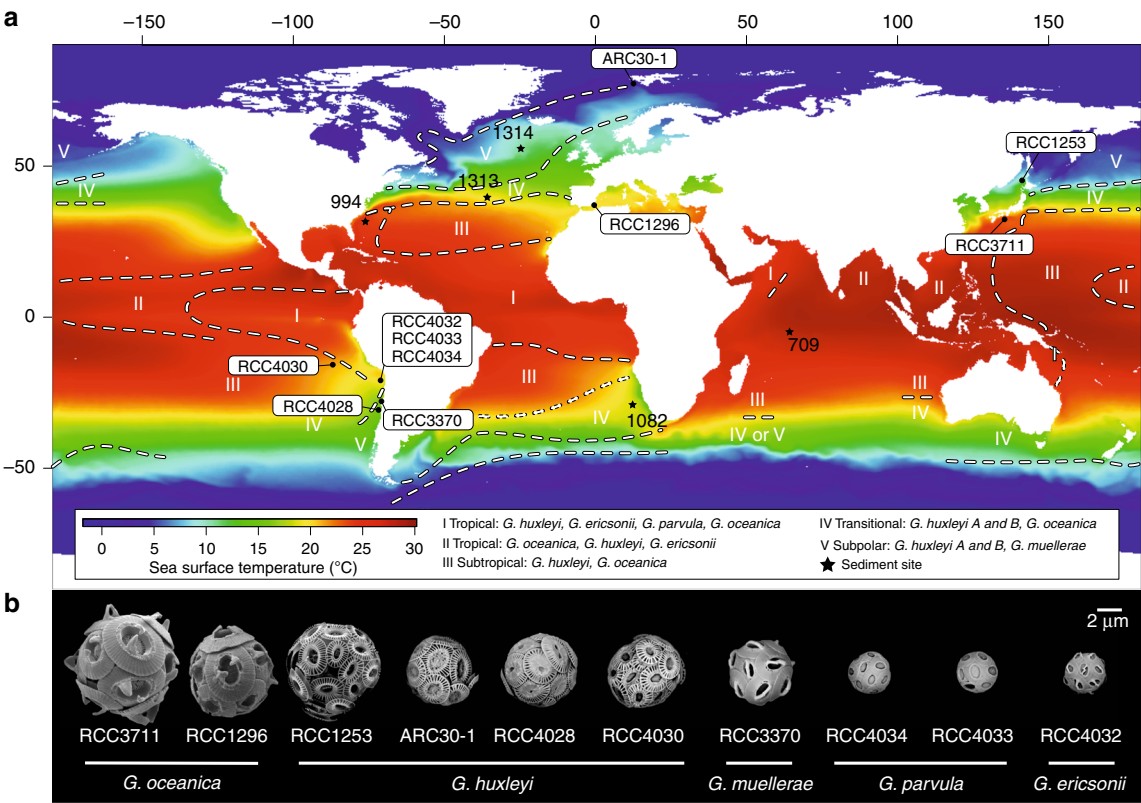

**Fig. 1** Global distribution of extant *Gephyrocapsa*. **a** Distribution of the *Gephyrocapsa* isolates and their corresponding flora (adapted from ref. [17] and updated with data from ref. [20]) shown over modelled-satellite SST monthly climatology from Modis Aqua (2002–2017) plotted with SeaDAS. Relative range of *Gephyrocapsa* flora are denoted with roman numbers. In the legend, species names are ordered by decreasing relative abundance in each assemblage. Sediment sites used in this study are also shown: 709[39], 994[87], 1082[88], 1313[89] and 1314[89]. **b** Scanning electron micrographs of the *Gephyrocapsa* strains used in this study (Scale = 2 μm)

11.6% of the 6278 trees analysed, significantly ($P < 0.05$) rejected the species tree. This type of incongruence can be caused by phenomena, such as incomplete lineage sorting (ILS) and is very common in genomes of different organisms[34]. The extent of phylogenetic incongruence caused by ILS is proportional to the effective population size of the ancestral species[34] and can, therefore, be used to estimate the population size of species that are extinct[35]. This allowed us to study the demographic history of speciation throughout the *Gephyrocapsa* phylogeny.

**How do new plankton species evolve**. The genome sequence data for 10 *Gephyrocapsa* strains allowed us to study evolutionary genetic processes underpinning speciation in this genus. For this purpose, we used Bayesian inference under the multispecies coalescent model that estimates effective population sizes ($N_e$) at every node in the phylogeny while taking into account the possibility of interspecific gene flow[36]. The results of this analysis are summarised in Fig. 2c, with the sizes of the circles proportional to estimated population sizes of extant and ancestral *Gephyrocapsa* species. Importantly, these genetic diversity-based estimates of *Gephyrocapsa* population sizes for extant species are congruent with current species abundance. In particular, *G. huxleyi*, which has been globally abundant since the beginning of the Holocene and became as abundant as other *Gephyrocapsa* species 70 ka[15], is now the most common coccolithophore in modern oceans (with a population size of the order of $10^{22}$ cells[37]), which is consistent with the large population size inferred for this species compared to other extant *Gephyrocapsa* (Fig. 2c; Supplementary Fig. 5 and Supplementary Table 2). Furthermore, the fossil-based estimates

of absolute and relative species abundance in this genus through time (Fig. 2d, e) are broadly congruent with our evolutionary genetic inference of population sizes in the ancestral *Gephyrocapsa* species (Fig. 2c). In particular, the common ancestor of all extant *Gephyrocapsa* species (node 1) and the common ancestor of all *Gephyrocapsa* species except *G. oceanica* (node 2) had the largest population sizes, which is congruent with particularly high *Gephyrocapsa* coccolith counts in sediments around 550–300 Ka (Fig. 2e). The origin of *G. huxleyi* around 300 Ka, that is well documented in the fossil record[15], apparently occurred in a very large population (node 2; Fig. 2c) without a significant speciation bottleneck (i.e. without a period of reduced population size due to origination of a species in a very small population). The highly distinctive phenotype of *G. huxleyi* (Fig. 1b) likely evolved throughout the range of the ancestral *Gephyrocapsa* species, as suggested by global synchrony of this event in the sediment record.

Our analyses indicate that interspecific hybridisation and low intensity gene flow has commonly occurred between *Gephyrocapsa* species throughout their evolutionary history. The estimates of interspecific gene flow, based on Bayesian inference under the multispecies coalescent model[36], are summarised in Fig. 2c as arrows, revealing gene flow between most nodes in the phylogeny. This is consistent with results of a Patterson's *D*-statistic test[38] that detected gene flow between *G. huxleyi* and *G. muellerae* and between *G. huxleyi* and *G. ericsonii/parvula* (Supplementary Fig. 6; Supplementary Tables 3 and 4). Thus, closely-related *Gephyrocapsa* species are not completely reproductively isolated and they co-occur to occasionally form hybrids. This finding rules out a purely allopatric mode of speciation and

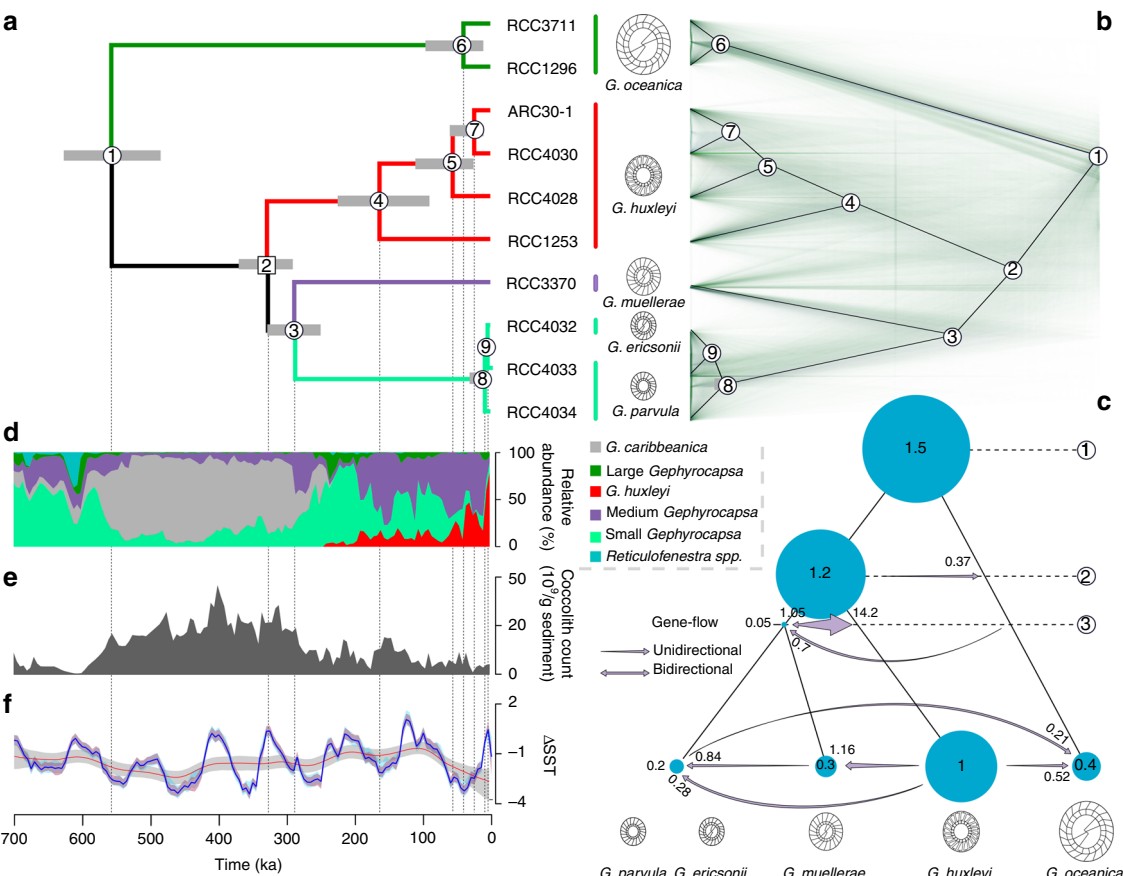

**Fig. 2** Evolution of extant *Gephyrocapsa* species. **a** Phylogenetic chronogram of extant *Gephyrocapsa* based on analysis of genome sequence data of 10 strains. Every node in the phylogeny has a posterior probability at 1. The dating of ancestral nodes is based on relaxed molecular clock calibrated with the first appearance of *G. huxleyi* (node 2; uniform prior 290–350 ka) in the fossil record. 95% Highest posterior density (HPD) intervals for ages are shown as grey bars. **b** Densitree plot based on 6278 phylogenies constructed for genomic fragments 10 kb long, and consensus species tree of extant *Gephyrocapsa* inferred from a multi coalescent method. All nodes are supported with 100% bootstrap. **c** Demographic modelling of *Gephyrocapsa* speciation history. The sizes of blue circles are proportional to estimated population sizes of extant and ancestral *Gephyrocapsa* species; the numbers inside blue circles ($=\theta/\theta_{Ghux}$) show population sizes relative to that in extant *G. huxleyi*. Direction of inferred interspecific gene flow is shown with purple arrows, with their widths and the number next to arrowhead showing gene flow intensity per generation. For estimates with confidence intervals see Supplementary Tables 2 and 3. **d** Stacked relative abundance of the Noëlaerhabdaceae assemblage including *Gephyrocapsa* and *Reticulofenestra* species over the last ~700 ka at site 1082[88] (Large *Gephyrocapsa* include *G. oceanica*; Medium, *G. margereli* and *G. muellerae*; Small, *G. aperta* and *G. ericsonii*). **e** Absolute number of the same assemblage in site 1082. **f** Global Δ sea surface temperature (ΔSST) over 700 ka[52], blue line is for mean variations, red and blue bands for 95% confidence intervals (respectively variability and jackknife). Red line corresponds to loess smoothing and associated grey band to 95% confidence interval

indicates that new *Gephyrocapsa* species formed in parapatry or sympatry, which is consistent with the frequent co-occurrence of different *Gephyrocapsa* species in the water column and in marine sediments.

**Repeated species radiations in *Gephyrocapsa* fossil record.** Using morphometric data from Matsuoka & Okada[39,40] we analysed the evolution of *Gephyrocapsa* coccolith morphology over the Pleistocene (1.8 Ma; Fig. 3a) measuring coccolith size, bridge angle and central opening (Fig. 3b; Supplementary Table 5) in successive populations with a sampling resolution of about 40 ka. The study revealed 3 pulses of coccolith size increase punctuated by episodes of size variance reduction. For convenience, we refer to these events of increasing size of coccoliths as 'Matsuoka–Okada cycles' (MO-cycles), and label them from oldest to most recent as MO1, MO2 and MO3 (Fig. 3a, Supplementary Figs. 7 and 8). Before the Pleistocene, *Gephyrocapsa* were typically rare and small (coccoliths <3 μm) and they become more abundant and larger (coccoliths >4 μm) from about 1.7–1.2 Ma

(MO1, Fig. 3a). This episode of coccolith size increase at 1.7 Ma was followed by a gradational process whereby distinct population clusters were recognised, with the two larger forms (species A and B in Matsuoka & Okada's Fig. 2[39]) disappearing at the same time. Only small *Gephyrocapsa* (<3 μm) were then present for about 250 ka (about 1.2–0.95 Ma). This event has been used for over 40 years as a key datum in biostratigraphic studies including ODP and IODP scientific ocean drilling projects[15]. This cycle was followed by two more sequences of gradual size increase (MO2 and MO3, Fig. 3a), initiated by an abrupt increase around 950 ka and separated by a shorter interval of smaller specimens only, apparent through the reduction of coccolith size variance (top-down, bottom-up) at around 0.5 ka. The available evidence indicates that most size reduction events were abrupt and globally synchronous, whilst size increases were gradational (Fig. 3a).

It remains unclear whether MO-cycles in *Gephyrocapsa* (Fig. 3a) and similar size pulsing in other plankton[23–25,41] represent species radiations followed by extinctions, or merely fluctuations in relative abundance of the same large- and small-sized species. In order to test this, we estimated the ages of

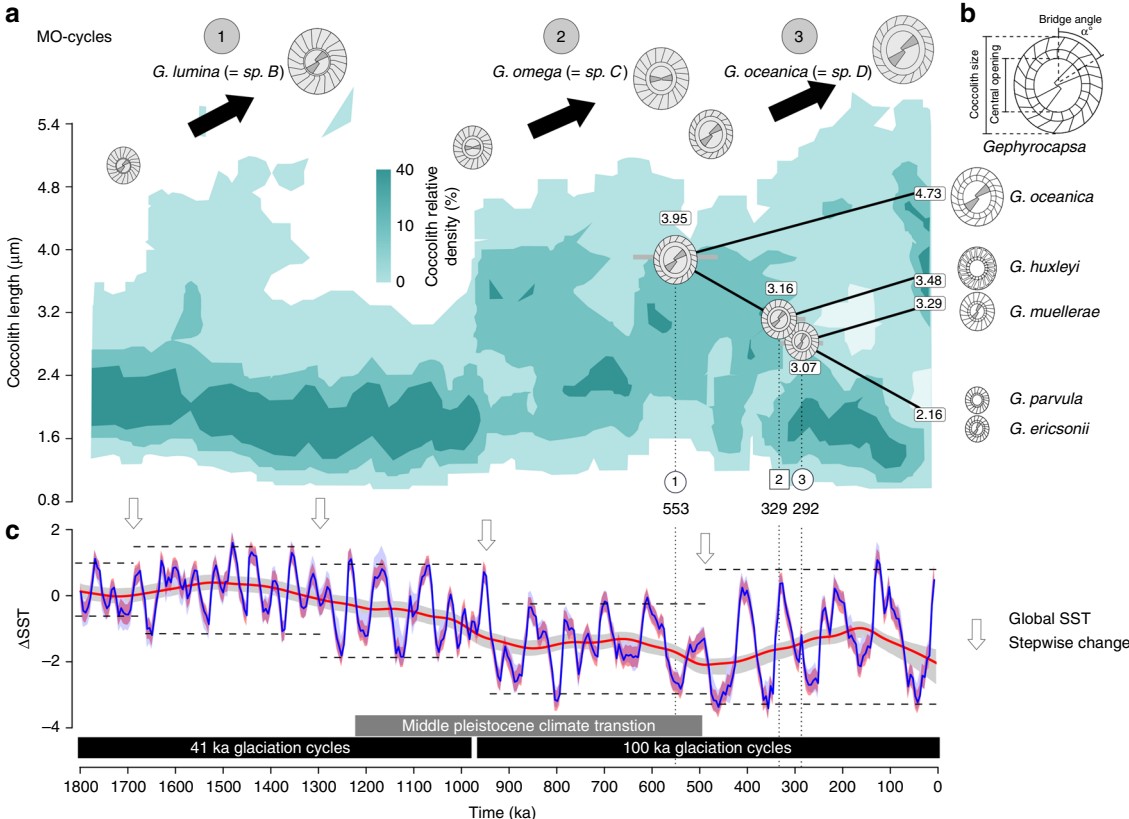

**Fig. 3** Matsuoka–Okada cycles and diversification in *Gephyrocapsa*. **a** Size variation of coccoliths in *Gephyrocapsa spp*. in the last 1.8 Ma at site 709[17,39], with three coccolith size enlargement events (MO1–MO3) associated with fossil species. Shades of green colour correspond to the relative density of coccolith. A phylogenetic chronogram of extant *Gephyrocapsa* based on relaxed molecular clock and calibrated with the first appearance of *G. huxleyi* (node 2) in the fossil record. Branches of the tree are scaled to coccolith length, estimated by ancestral trait reconstruction (Supplementary Fig. 9), and shown for each node. Mean coccolith length (in μm) for modern and ancestral species is shown in white rectangle at each tip and node. **b** Coccolith traits measured. **c** Global Δ sea surface temperature (ΔSST) over 1800 ka[52]. Blue line is for mean variations, red and blue bands for 95% confidence intervals (respectively, variability and jackknife). Red line corresponds to loess smoothing and its associated grey band to 95% confidence interval. White arrows correspond to stepwise changes in relative global SST

speciation events in the *Gephyrocapsa* genus from our newly generated genome sequence data. We used a relaxed molecular clock[42] with the calibration point corresponding to the most reliable fossil date, the first appearance of *G. huxleyi*[28] around 300 ka (node 2 on Fig. 2a). This allowed us to date other nodes in the phylogeny, revealing that the most recent common ancestor (MRCA) of extant *Gephyrocapsa* species arose around 553 ka (node 1 on Fig. 2a and Supplementary Table 6). If MO-cycles were caused by the change in abundance of the same large- and small-sized species through time, then the MRCA of extant *Gephyrocapsa* species would be expected to be several million years old (coincident with first appearance of the *Gephyrocapsa* ~3.5 Ma). On the other hand, if each MO-cycle represents an independent burst of species diversification into larger and smaller forms, followed by extinction, then we would expect the MRCA to be much younger—around 0.5 Ma—the age of the last MO-cycle. Our estimate of the *Gephyrocapsa* MRCA age ~553 ka clearly demonstrates that all extant diversity in this genus evolved within the most recent MO-cycle 3, which indicates that MO-cycles likely represent consecutive species radiations, followed by globally synchronous extinctions. This is also supported by the detailed analysis of coccolith morphology. In particular, the comparison of larger coccoliths from different MO-cycles revealed that they consistently differed from each other in characteristics such as bridge angle and central opening size (Supplementary Fig. 1 and Supplementary Table 7), indicating

that different MO-cycles were caused by the spread of genetically distinct species: consecutively *G. lumina* Burky, *G. omega* Burky and *G. oceanica* Kamptner.

Evolution of extant *Gephyrocapsa* diversity within the last MO-cycle 3 involved significant morphological diversification (Figs. 1b and 3a), including evolution of a wide range of coccolith sizes and degrees of calcification and at least two independent events of bridge loss, one corresponding to the establishment of *G. huxleyi* and the other to *G. parvula* (Fig. 1b; Supplementary Fig. 9). This resolves the ambiguity of whether the bridge was gained or lost through time and illustrates the evolutionary dynamic of this diagnostic character[20]. This evolutionary pattern implies that more non-bridged members of the Noëlaerhabdaceae might have derived from bridged forms throughout the evolution of *Gephyrocapsa*, with further instances of bridge loss likely having occurred in previous MO-cycles. Furthermore, the example of *G. ericsonii* and *G. parvula* also suggests that other co-occurring bridged and non-bridged *Gephyrocapsa* could have been intergrading in the past.

Within each MO-cycle, there is a pattern of gradual coccolith size increase, followed by abrupt decrease. It has been suggested that such gradual increase in size, often referred to as Cope's rule[21,22], is a passive phenomenon caused by a lower physiological limit in size[43,44]. Therefore, after each extinction event, when primarily the smaller less specialized taxa survive[45], the maximal size in a group gradually drifts upwards, while the

median and the mode of the size remain low, as can be seen in Fig. 3a, where the darker colour indicates that most of the coccoliths remain in the smaller part of the size spectrum, while the upper limit increases. However, Heim et al.[22], who tested Cope's rule on the fossil record of marine organisms, demonstrated that the increase in size could not be explained by neutral drift and thus is possibly driven by selection for larger size. Our molecular genetic data indicate that the most recent example of Cope's rule in *Gephyrocapsa* is actually indicative of a much larger evolutionary turnover than was suspected previously, with the size increase is simply the most visible expression of a major species radiation underpinning the MO-cycle.

Further back in the fossil record, events similar to MO-cycles have punctuated the evolution of *Gephyrocapsa* ancestors over at least the past 15 Ma[46,47]. Other plankton groups including the diatoms[23], dinoflagellates[24] and planktic foraminifera[25] all demonstrate patterns of macroevolutionary variation in size attributed to changing upper ocean structure over similar timescales. Finer scale evolutionary events in the *Gephyrocapsa* ancestry, with characteristics similar to MO-cycles, have also been identified, such as periods of dramatic size reduction[43]. Furthermore, the history of the coccolithophore genus *Calcidiscus* displays the repeated development of coccoliths similar in appearance but at stratigraphically distant intervals[41]. These observations indicate that pulsed events in plankton evolution may be widespread across the coccolithophores and our results indicate that they represent species radiations separated by abrupt extinctions.

**Possible drivers of repeated plankton species radiations**. MO-like macroevolutionary patterns in a range of plankton groups have been inferred to relate to changes in environmental conditions[24,43]. Temperature fluctuations are likely to selectively impact abiotic and biotic processes[48]. In the open ocean and the photic zone, variation in the thermocline depth and/or strength is well known to affect phytoplankton community structure through changes in the availability of light and nutrients for growth, while modifying available niches[49]. In removing the dominance of one keystone species, climate change may promote opportunistic responses of closely-related co-occurring species bearing relative adaptive advantages[50].

MO events (Fig. 3a) are too extended to follow the periodic Milankovitch glacial cycles of the Pleistocene (Fig. 3b), but the duration and pacing of the MO events seem to coincide with phases of different maximum global ice volume expressed as periods of increased amplitude in oceanic $\delta^{18}O$[51], and alkenone-derived SST[52] cycles towards the modern day (Supplementary Fig. 8). Northern hemisphere glaciation intensified ~2.58 Ma[53], long before MO1 cycle, and these ice sheets oscillated at a regular 41 kyr period spanning from 2.58 to ~0.95 Ma (Fig. 3b). MO1 started after a global cooling event in SST around 1.8 Ma (event occurring from 2.1 to 1.8 Ma[54,55]) and ended abruptly ~1.2 Ma with the onset of the Mid-Pleistocene climate Transition (MPT; ~1.2 Ma to ~500 Ma when climate evolved from being dominated by a 41 ka to a 100 ka periodicity)[56]. There was then a period of ~250 kyrs of apparent stasis in coccolith size followed by a documented abrupt increase in ice volume and increased ocean storage of carbon[51,57] around 950 ka, which coincided with the upturn in coccolith size during the MO2 cycle. The subsequent extinction and reduction in coccolith size of the MO2 cycle coincided with the end of the MPT ~500 ka when the glacial cycles in climate first attained their greatest amplitude and a regular 100 kyr periodicity. The extinction that triggered the start of MO3 also coincided with the most northerly migration of the subtropical front to impinge on southern hemisphere

continents[58] and potentially restrict flow of low latitude waters between ocean basins. If oceanic fronts do act as barriers to gene flow[59–61], then this restriction could have isolated the low latitude basins making endemic populations prone to higher extinction rates. MO3 then proceeded during the regular 100 kyr fluctuations.

Other plankton also appear to evolve in response to step changes in global cooling. Evolution of diatoms in the Southern Ocean has shown diversification rates punctuated by several instances of massive extinctions closely related to global stepwise cooling suggesting a sensitivity to long-term climatic events[62]. Beyond the marine realm, the evolution of plants and animals also show climatically-sensitive evolutionary trends that are well characterised in hominids, Bovidae, birds and plants[63]. Although the mechanisms may be different in different realms, the evolutionary pulsed turnover of new species[63] may generally be associated with step changes in environmental conditions.

**Concluding remarks**. This study is the first to combine the analysis of plankton fossil data with evolutionary genetic analysis of whole-genome sequences to advance our understanding of plankton speciation. Our study indicated that pulses of coccolith (and likely cell[17,64]) size change observed in the plankton fossil record are caused by repeated species radiations rather than fluctuations in the relative abundance of large- and small-celled species. These species radiations comprise rapid diversification of coccolith sizes and morphology, followed by abrupt globally synchronous extinctions recorded in ocean sediments every ~0.5 million years. Similar patterns of cycles in size appear to be common in many plankton groups[43] and may also represent repeated pulses of species radiations and extinctions that lead to high turnover in diversity. Our evolutionary genetic analyses of coccolithophore genomes reveal the mode of speciation during species radiations of marine phytoplankton. In particular, our results rule out purely allopatric evolution in *Gephyrocapsa* and indicate that new species in this genus have formed in sympatry or parapatry, with occasional hybridisation between the species.

These results help to build a link between macroevolutionary patterns observed in the fossil record and micro-evolutionary processes underlying new species formation in marine micro-plankton. The combined analysis of fossil and evolutionary genomic sequence data is a major step forward from fossil/sediment-only or sequence-based only studies. Further work with continuous records of sufficient resolution combined with evolutionary genomic sequence data, could resolve whether pulsed radiations are a fundamental characteristic of macro-evolution both terrestrial and marine. Establishing the prevalence of such cycles is the first step to understanding the macroevolutionary process. The integration of population and evolutionary genetics with diverse fossil and climatic records could reveal much further insight into the mechanistic coevolution of biodiversity, productivity, the carbon cycle and climate.

## Methods

**Origin and morphological characterisation of strains**. Clonal Noëlaerhabda-ceaen strains (Supplementary Table 1) from the Roscoff Culture Collection (RCC; roscoff-culture-collection.org) were maintained in K/2(-Si,-Tris,-Cu) medium[65] at 17 °C with 50 μmol-photons.m$^{-2}$ s$^{-1}$ illumination provided by daylight neon tubes with a 14:10 h L:D cycle. Calcified cells were harvested at early exponential growth phase on 0.22 μm nucleopore filters, then dried in a 55 °C oven for 2 h. Following gold coating, filters were observed with a Phenom scanning electron microscope. Morphometric measurements (Supplementary Table 5) were carried out according to Bollmann[66] with a minimum of 60 isolated coccoliths per strain using ImageJ software (http://imagej.nih.gov/ij/).

**Stratophenetic data**. Our reconstruction of the fossil record of *Gephyrocapsa* is based on the study of Okada and Matsuoka[39]. Morphometric data from this study, including measurements of coccolith size, bridge angle and central opening

(Fig. 3b; Supplementary Table 7) over the Pleistocene were provided by Prof Matsuoka (Kochi University, Japan) and is available from Pangaea database (https://doi.pangaea.de/10.1594/PANGAEA.903745). This data were reanalysed and re-interpreted in the context of modern taxonomic concepts of the species represented. The data was based on measurement of transmission electron microscope images of carbon-replicated coccoliths from sediments collected from the tropical Indian Ocean during Ocean Drilling Project Leg 115[39].

**DNA extraction.** Cell cultures were harvested by centrifugation ($4500 \times g$, 15 min), washed twice with TE buffer, and suspended in 10 ml of lysis buffer (Tris, 0.1 M; EDTA, 0.05 M; NaCl, 0.1 M; 1% SDS; 2% N-lauroylsarcosine, proteinase K 200 mg/mL, pH 8.0) and incubated at 55 °C for 2 h for extraction of total genomic DNA. DNA was then purified with equal volumes of phenol and chloroform and precipitated with ethanol[67]. For each sample, quantifications of nucleic acids were performed either with a Qubit 3.0 fluorometer (Thermofisher Scientific, Inc.) or a Nanodrop. DNA extracts were sent to the Wellcome Trust Centre for Human Genomics, Oxford (WTCHG) for sequencing. Paired-end libraries were prepared individually, barcoded, and then combined prior to sequencing. Libraries were sequenced using an Illumina HiSeq 2500 sequencing platform to produce 150 base-pair (bp) paired-end reads. The amount of raw data generated for each strain is listed in Supplementary Table 1. All sequences are available from NCBI under bioproject number PRJNA532411.

**Mapping of reads.** After quality trimming with Trimmomatic[68], the sequence reads from *Gephyrocapsa* strains were mapped to the *G. huxleyi* CCMP1516 reference genome[69] with BWA-MEM[70]. Despite low sequence divergence (<3% total sites) between the strains analysed, the proportion of reads mapped to reference was relatively low (40–69%; Supplementary Table 1) likely due to the known variability of *G. huxleyi* pan-genome with a large proportion of the genome missing in many strains[69]. Duplicated reads were removed using Picard (https://broadinstitute.github.io/picard/). The Genome Analyses Toolkit (GATK)[71] was then used for base quality recalibration, local realignment around Indels (insertions/deletions), and SNPs (Single Nucleotide Polymorphisms) calling. For downstream analyses, obtained SNPs in vcf file format were converted into fasta multiple alignment sequences per contig (7795 contigs) using a custom script *vcf2fas* (available at https://github.com/brunonevado/vcf2fas).

**Phylogenetic inference.** For phylogenetic analysis of the Noëlaerhabdaceae, alignments of each contig with at least one missing individual were discarded reducing the dataset from 7795 to 2137 contigs, hence shrinking the total length of the alignment from 167.7 to 154.7 megabases, accounting for 4,105,458 positions which were phylogenetically informative. Initially, we conducted a species tree reconstruction using a concatenation-based approach, by first concatenating all 2137 contigs. We then performed a maximum likelihood (ML) tree inference on this concatenated alignment in RAxML 8[72] using the GTRGAMMA model and 100 bootstrap replicates. This approach allows production of a bootstrapped species tree with branch lengths. Next, we reconstructed a species phylogeny using a multispecies coalescent-based approach to account for incomplete lineage sorting (ILS). Contig-alignments were split in 13,810 separate regions 10 kb long, of which 6278 were retained for phylogenetic analyses after excluding the alignment positions with >20% of gaps or missing data. For each of these 6,278 alignments, we performed a phylogenetic reconstruction using the GTRGAMMA model and 100 bootstrap replicates in RAxML. Best ML trees with bootstrap replicates were then used to produce a species tree using ASTRAL[73] with 100 bootstrap replicates using both site-wise and gene-wise resampling. The same ASTRAL analysis was repeated with shorter (5 kb long) windows, which yielded the same species tree as the analysis with 10 kb long windows (Supplementary Fig. 4).

An Isochrysidales tree was reconstructed in order to provide a root to the Noëlaerhabdaceae phylogeny by using two outgroups: *Tisochrysis lutea* and *Isochrysis galbana*, which are the closest available relatives of *Gephyrocapsa*. The genome sequence for *T. lutea* strain CCAP926/14 was obtained from http://www.seanoe.org/data/00361/47171/[74], while the transcriptome sequence of *Isochrysis galbana* strain CCMP1323 was obtained from the Marine Microbial Transcriptome Sequencing Project[75]. To identify orthologs we extracted coding sequences (CDS) from the published *G. huxleyi* genome[23] and compared them with the outgroup sequences using Orthofinder[76] v2.2.6 with standard parameters. The corresponding regions were extracted from *Gephyrocapsa* alignments, translated to predicted protein sequence and aligned with outgroup sequences using MAFFT[77]. The resulting alignments were filtered with alignment columns with >20% missing data excluded before concatenating all genes. We then performed a maximum likelihood tree inference on this concatenated alignment with RAxML using the PROTGAMMAAUTO model and 100 bootstrap replicates. To assess the reliability of the position of the root we also used the more complex mixture model CAT-GTR. The position of the root for the Noëlaerhabdaceae phylogeny on the branch connecting *G. oceanica* with other *Gephyrocapsa* species was consistent regardless of the outgroup and model used (Supplementary Figs. 2 and 3).

**Molecular clock analyses.** In order to better interpret patterns of speciation and morphological changes, we estimated divergence times for the *Gephyrocapsa* using uncorrelated and autocorrelated relaxed-clock approaches implemented in the mcmctree program[42] from the PAML 4.9b package[78]. As mcmctree is able to incorporate multiple loci, we based this analysis on the 6278 genomic sequence alignments 10 kb long obtained as described above. Alignment columns with missing data, gaps and ambiguity characters were excluded from analysis (mcmctree setting *cleandata* = 0). A uniform prior with min = 290 Ka and max = 350 Ka was used as a calibration time for the divergence between *G. huxleyi* and other *Gephyrocapsa* (node 2 on Fig. 2a), based on first occurrence of *G. huxleyi* in the fossil record 291 Ka[28], which is the only confident calibration point available in the fossil record of the group. As the mcmctree program requires definition of a "safe maximum" for the age of the root, we set this parameter to "<10" Ma. The analysis was conducted with an autocorrelated relaxed molecular clock (*clock* = 3) under HKY85 + G5 model (*model* = 4; *alpha* = 0.5) and full likelihood option (*data* = 1) in mcmctree and comprised two runs of 500,000 MCMC steps after a 50,000 burn-in. To test the robustness of the age estimates to the parameter values and the type of relaxed molecular clock used, we also conducted multiple shorter (200,000 MCMC steps after a 20,000 burn-in) runs on a subset of the 6278 genomic sequence alignments that included 500 randomly selected alignments. This subset was used for running the mcmctree program with both the independent rates model (*clock* = 2) and the correlated rates model (*clock* = 3) under HKY85+G5 model with full likelihood option, which yielded similar results (Supplementary Table 6). Posterior time estimates were shown to be sensitive to the prior for the distribution of divergence time across the loci (parameter *rgene_gamma* in mcmctree)[79]. Thus, we ran the program with fairly diffuse Dirichlet-gamma prior for the mean substitution rate (*rgene_gamma* with shape parameter, α = 1) and different values of β parameter ranging from 10 to 500, which yielded consistent time estimates (Supplementary Table 6). The convergence of the runs was monitored using Tracer[80].

**Phylogenetic discordance.** A DensiTree plot was produced using Densitree[81] version 2.2.1 to visualise phylogenetic discordances between loci based on 6278 ML trees reconstructed for 10 kb long genomic regions. For each tree retrieved from these 10 kb fragments, we used the *pruneTree* function in the R *phangorn* package[82] to collapse nodes with bootstrap support <75%. Trees with no nodes over 75% bootstrap support were discarded. Using the *root* function, each of the pruned trees was then rooted by *G. oceanica*, as the protein concatenated tree suggested, and each tree was made ultrametric using the *chronos* function with default settings in the R *ape* package[83]. Resulting trees were then loaded into DensiTree[81]. DensiTree plots were produced using the consensus trees produced by DensiTree (in which branch lengths are averaged across all trees for a given topology) with the following settings (star tree, consensus *width* = 1, consensus intensity 28.1, and default values for all other settings).

To determine how much of the observed variation among the 10 kb fragment trees was due to genuine incongruence, rather than simply lack of phylogenetic signal, we used Shimodaira–Hasegawa (SH) tests[33] implemented in CONSEL version 0.2 on all 10 kb alignments. The procedure first uses the phylogenetic inference program PhyML version 3.0[84] over two runs for each locus. The first run uses an unconstrained topology, and the second run constrains the topology to that of the species tree. Both runs were performed using the GTR substitution model without bootstrap replicates. Site-likelihoods from these runs were then compared in CONSEL and FDR correction[85] was applied to the *P* values for each SH test. Tests for which the FDR corrected SH-test *P* value (*Q* value) for the constrained tree is <0.05 were considered to significantly reject that species tree.

**Analysis of demographic history.** The analysis of demographic history, including divergence times, effective population sizes, and gene flow, was conducted with the Generalized Phylogenetic Coalescent Sampler (G-PhoCS) program 1.2.3[36]. Following the philosophy of Bpp[86], G-PhoCS is based on a full coalescent isolation-with-migration model and allows asymmetric gene flow between taxa. This approach estimates effective population size ($N_e$) at every node in the phylogeny as a compound parameter ($\theta = 2N_e\mu$) with mutation rate ($\mu$). This $\mu$ is not known for *Gephyrocapsa* but cancels out in the comparisons (ratios of $\theta$) between different species; thus, we normalised all population sizes in the *Gephyrocapsa* phylogeny by modern *G. huxleyi* size, which is known to be astronomically large as this species is ubiquitous and abundant in modern oceans. Gene flow was assessed using "migration bands", as implemented in G-PhoCS, which estimate per generation migration rates ($m_{st} = M_{st}/\mu$), where $M_{st}$ is the proportion of individuals in target population $t$ that arrived from source population $s$. G-PhoCS uses haplotype data with no filtering for minor allele frequency to model past species demography. For this analysis we used 1000 randomly selected 1 kb long regions throughout the genome; alignments of sequences of these regions from the 10 sequenced *Gephyrocapsa* strains were used to run G-PhoCS with 1,000,000 MCMC steps with 100,000 burn-in. Runs were visualised using Tracer v1.6 program[80] to ensure convergence.

As the hypothesis of introgression between *Gephyrocapsa* species was formulated in an earlier study based on cytoplasmic markers, introgression between species was also investigated as a cause of the observed phylogenetic discordances[20]. We performed a Patterson's *D*-statistic test[38] which compares two phylogenetically incongruent site patterns of ancestral (A) and derived (B) alleles ABBA—(((A,B),B),A) and BABA—(((B,A),B),A) on a four-taxon phylogeny with

the topology: (((P1,P2),P3),Outgroup). If the incongruence is due to ILS, the frequencies of these site patterns are expected to be equal, but in the case of introgression between P3 and either P1 or P2, they are expected to be biased toward the site pattern that clusters the introgressed taxa together. Block Jackknifing (with each locus representing a single block in the context of our dataset) was then used to determine significance. We used a custom ABBA/BABA script (available at https://github.com/brunonevado/calcD_from_fas) to test every phylogenetically congruent three-species subtree using *G. oceanica* as outgroup.

**Reporting summary**. Further information on research design is available in the Nature Research Reporting Summary linked to this article.

## Data availability

All genome sequencing data generated in this study have been deposited in the National Center for Biotechnology Information (NCBI; https://www.ncbi.nlm.nih.gov) and are accessible under bioproject number PRJNA532411. Two previously published *G. huxleyi* genomes (RCC4028 and RCC4030) that were used in this study are available from NCBI under accession numbers ERR695589 and ERR695590. Coccolith morphometric data analysed in this study is available from Pangaea database (https://doi.pangaea.de/10.1594/PANGAEA.903745).

## Code availability

Custom programs and scripts are available at GitHub: https://github.com/brunonevado/.

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

## Acknowledgements

This work was supported by a grant from the John Fell Fund (Grant 152/079) to D.A.F. and R.E.M.R. and B.B.S.R.C. grant (BB/P009808/1) to D.A.F. The authors thank Prof Matsuoka (Kochi University, Japan) and Prof Baumann (University of Bremen, Germany) for providing Gephyrocapsa coccolith fossil data and the staff at the WTCHG (Oxford) for high throughput sequencing and initial data processing.

## Author contributions

D.A.F. and R.E.M.R. conceived the project. E.M.B. and E.L.Y.W. prepared the samples for sequencing. E.M.B. and I.P. conducted SEM analyses. E.M.B., B.N. and D.A.F. conducted the analyses of sequence data. E.M.B., K.H. and J.R.Y. analysed and interpreted strato-phenetic data. D.A.F. and E.M.B. wrote the paper with contributions from R.E.M.R. All authors contributed to editing the paper.

## Additional information

**Competing interests:** The authors declare no competing interests.

