## [Peer Review File · Nature Communications]

Reviewers' Comments:

Reviewer #1:

Remarks to the Author:

This is a very well-written manuscript that is the first to combine the analysis of fossil coccolith data with genomic analysis of extant species to advance the understanding of plankton species radiation. It is novel and I pretty much appreciated the effort of the authors introducing such a detailed data-set including morphologic analysis, molecular phylogenetic work, comparison of the phylogenetic evolution to size variations of *Gephyrocapsa* coccoliths as well as a demographic modelling of *Gephyrocapsa* speciation history that provide strong evidence for its conclusions.

I see no problems in the work, which I like very much, and I believe that the integration of population and evolutionary genetics with diverse fossil records sheds novel light on the speciation mode in the recent evolution of a key phytoplankton group. I can follow essentially all the conclusions drawn from this study well, so that I recommend the manuscript for publication with minor revision only.

I have noted only minor suggestions and typos:

Titel - Since the study relies exclusively on the coccolith genus *Gephyrocapsa*, it should, in my view, also find expression in the title of the paper.

Line 51ff - The sentence "Coccolith sediments represent a rich source of nannofossils ..." reads a bit strange and should be rephrased. Furthermore, the authors should be aware that although many occurrence and extinction dates may appear to be globally synchronous, there are quite a number of examples where such an event is recorded in different stratigraphic positions in different oceanic areas (see Raffi et al., 2006 QSR).

Line 105ff - I wonder how the authors assess the morphologic study of *Gephyrocapsa* by Bollmann (1997) and the distinction of morphological groups within the genus characteristic of different environments in the context of their findings!

Line 138 - It is not entirely true that "*G. huxleyi* is the most common coccolithophore ... for at least 70 ka"! Actually it is of global dominance since the beginning of the Holocene, but it is as abundant as other *Gephyrocapsa* species since 70 ka, as you can also see in Figure 2d.

Line 185ff - The species radiation in *Gephyrocapsa* fossil record is reduced to variations in coccolith size. I can understand that it is necessary to break down such a development on individual characters, and I do not want to criticize it, but are there similar variations in other parameters? The species are not only defined by the size but also, according to Bollmann (1997), by the bridge angle.

Line 260ff - I am not fully convinced that sea-surface temperature variations (alone) are likely to drive the species radiations and extinctions. I see a connection to the climate evolution but not necessary a causal relation (and doubt that this is possible to infer from the Δ SST).

Line 306 - I think it is not mentioned before how cell size and coccolith size are related? What results in the statement that changes in cell size are observed?

Line 325 - I am a bit puzzled that the paper ends with a lot of questions for future research.

Line 344 - Its Stratophenic data ...

Table 1 - its not *G. margerelii* but *G. margereli* (although both spellings are also used in Nannotax)

Supplementary Figure S2 – I would recommend adding the names of the four Gephyrocapsa species here.

Lines 677, 685 – Gephyrocapsa spp (spp not in italics)

Reviewer #2:

Remarks to the Author:

The manuscript by Bendif et al. presents a study on the diversification patterns of haptophytes of the genus Gephyrocapsa. Despite the ecological importance of these species, this study is particularly interesting because it reports a comprehensive analysis of genetic and paleontological data. Together, these two lines of evidence picture an scenario of repeated speciation/extinction cycles in the last 2 million years. The integrative nature of the methodology and the conclusions of this study are very relevant for the community of evolutionary biologist.

I really enjoyed reading this paper. It is very well written. The results are presented in a logical way and the methodology seems (mostly) robust. I only have two major concerns regarding the methodology: the molecular clock analyses and the rooting of Gephyrocapsa phylogeny with Tisochrysis.

Molecular clock. The methods of this section are very poorly described and key important settings are not mentioned. Which type of relaxed molecular clock was used, autocorrelated or uncorrelated? How was the calibration specified? Calibrations should always incorporate its inherent uncertainty (e.g. with a uniform prior with min and max ages). What was the assumed prior for the overall rate parameter (rgene_gamma)? This parameter has been shown to have a large effect on posterior time estimates (Dos Reis et al. 2014 Syst Bio 63). Were timetrees inferred with full or approximated likelihood? Please, specify all relevant parameters. In addition, because the divergence times are one of the key results on which the paper is built, it would be ideal if the authors could assess the robustness of their results using some sensitivity analyses, such as comparing the effect of using different clock models, different calibrations (if available), etc.

Rooting. Supplementary Fig. S1 suggest a very long branch separating Tisochrysis from Gephyrocapsa, which led me wonder if there could not be a "random rooting" effect. When the outgroup is very distant, its sequence likely contains insufficient information on the polarity of characters within the ingroup and can led to the long branch of the outgroup being connected in any branch in the ingroup with low reliability, "randomly" (Wheeler 1990 Cladistics 6; Hirase et al. 2016 Genome Biol Evol 8). If this were the case, changing the root of the Gephyrocapsa phylogeny would have a huge impact on molecular clock. This could be alleviated by using a closer outgroup (if available) or a broader sampling of outgroups.

In relation to this rooting problem, the presence of undetected paralogs in Tisochrysis might exacerbate the already long branch, leading to random rooting artifacts. The authors identify homologous sequences in Tisochrysis as best blast hits with the other species. However, this strategy alone is not enough to ensure that the sequences are orthologous and the authors do not report any step towards identifying and removing possible paralogs. It is known that the inclusion of a few paralogs can negatively affect phylogenomic inference and molecular clock analyses (e.g. Shen et al. 2017 Nat Ecol Evol; Siu-Ting et al. 2019 Syst Bio). The authors could exclude paralogs using a broader taxon sampling in the and using phylogenetic inference of locus trees, for example. Also, the use of more complex mixture models that often fit the data better (e.g. CATGTR or C60 family) could also help assess the reliability of the position of the root.

Other remarks:

The correspondence between extant *Gephyrocapsa* species and large/medium/small species from the cores (e.g. in Figure 2) is unclear, at least for me as a non-specialist in the system. For example, in Table 1, column "Remarks", this is particularly surprising: what does it mean that *G. lumina* is "sometimes referred to as large *Gephyrocapsa* and *G. oceanica*" etc.? Is it because determining morphospecies in fossils is difficult or because the taxonomy has been changing and so species names are not comparable across studies? It would be great if the authors could clarify this for non-specialists.

Similarly, several terms specific to coccolithophorids are mentioned in the paper (coccolith length and presence of bridge; lines 241-251). Could you please briefly explain what they are and what their relevance is?

Data and code availability. The datasets (including raw morphometric data) and code (including the pipeline for testing for topological incongruences, line 435) should ideally be made available as supplementary material or deposited in recognized repositories (e.g. dryad, figshare) to ensure reproducibility and avoid that this data might be lost for the scientific community.

Sampling locations. The localities where sediments and living strains were obtained are geographically disparate (Figure 1). Could this have affected the definition of morphospecies in either the sediments or the sequenced strains, or hinder some important local effect?

Methods: DNA extraction. How many cells were used for sequencing? Or can this be approximated somehow, e.g. dry weight?

Methods: read mapping. It would be great to have a supplementary table with details on the genome data, including mapping statistics per genome. Also, 40% read mapping might seem quite low (unacceptable in most animals and plant systems). I assume this is due to the known variability of *G. huxleyi* pan-genome (Read et al. 2013 Nature 499), is it correct? Again, this might not be obvious for the non-specialist and merits a brief explanation.

Methods: coalescent species tree. ASTRAL analysis are based on 10kb windows. These are quite large regions and ASTRAL assumes no recombination within each of these 10kb windows, which seems unlikely. Could this have affected the ASTRAL tree?

Figure 1. Add number and unit to the unit scale (to avoid being confused with the lines delimiting species).

Figure 2 caption. is ASST a global measurement or only of site 1082?

Supplementary Fig. S4. Replace "um" by the right symbol μ .

Supplementary Fig. S5. Rainbow color scheme is misleading for representing continuous measurements such as length and probability. A simple grayscale would do it. Also include species names (not only vouchers) and specify coding for bridge (I assume 1=presence and 0=absence).

Supplementary Table S4. The results from ABBA/BABA tests could be better represented graphically than in a table, e.g. using violin plots (see Irisarri et al. 2018 Nat Commun 9, Fig 3 <https://www.nature.com/articles/s41467-018-05479-9>).

Lines 61-62: this sounds like Cenozoic Reticulofenestra would be ancestors of Gephyrocapsa species, which makes no sense given that Reticulofenestra are also extant species. Could this be rephrased?

L87-88: How were morphospecies delimited? Was it based on a phylogenetic analysis of morphological data? If so, please present the tree in the supplement. Also, the underlying morphological data should be made available in a recognized repository. Or is this data in Supplementary Table S5? This table is also not mentioned in the text.

L124: incongruence patterns can also derive from other phenomena, including introgression, as it is revealed by the coalescence analyses by the authors themselves (Fig. 2c).

L235 (Fig 3 caption): aren't > are not

L268-270: the coincident patterns between the morphospecies successions and planetary cooling are not very obvious for me from Fig 2d-f.

L296: hominids should not be capitalized.

L306: I think a softer phrasing such as "Our study suggests" or similar would be better here. This conclusion depends on the credibility of the molecular clock analysis.

L325 Regarding: "Why does diversification lead to increased range of sizes?" Could it be possible that size increase is only one of the obvious patterns that we recognized as speciation?

L328: Regarding: "Are the cycles/extinctions driven by external factors such as climate/cooling as suggested here, or are the cycles internally self-limiting perhaps as a result of the increase in size and pressure on some limiting resource" ... or maybe both?

L355: either ... and

L374, 401, 692 and others: "maximum likelihood", "parsimony" or "incomplete lineage sorting" should not be capitalized. Also, introduce abbreviations on first time of appearance rather than in the methods section.

L380-381: Could you clarify what you mean with "after excluding the alignments with less than 80% of sites without missing data"? Were sites with >20% excluded or instead entire 10kb alignments when they had <80% missing data? Were gaps considered missing data?

L394: blastp

References. Format should be homogenized.

Reviewer #3:

Remarks to the Author:

The article entitled "Repeated species radiations in the recent evolution of a key marine phytoplankton lineage" presents a study of the evolution modus of phytoplankton. The authors have generated genome sequencing of 10 strains of the Gephyrocapsa genus and used these data to reconstruct the evolutionary history of the constitutive species. The author compare their results to the fossil record of

the coccolithophores and highlight a pattern of repeated species radiation that they designate as the "Matsuoka-Okada" (MO) cycles that seem to have a cyclicity of about 0.5 Ma.

I have appreciated to read the article. The author have generated a comprehensive genomic dataset, compared their result to the patterns of the fossil record, and showed many points of convergence.

The writing is clear, the illustrations of high quality and the author show clearly new patterns of Coccolithofores evolution. I have only critics on the last third of the article where the author are discussing what drives repeated plankton species radiations and extinction and my impression is that this section of the article is weaker. This is because the authors try to link paleoclimatic changes to the MO cycles but this does not constitute a solid argumentation in my opinion, and I think that solving these questions would require an independent study with statistical tool and comparison with parameters other than the SST. In addition, the author rule-out pure allopatric speciation as a mode of speciation in *Gephyrocapsa* but this does not sound to me as a groundbreaking discovery because, as the author mentioned it in their introduction, the lack of geographic barrier in the ocean alluded speciation to sympatric mode.

Similarly, they authors ask in their conclusion why diversification lead to an increase of size. This is, in my opinion, not a novel question and it has actually produced a prolific literature already. This phenomenon of increase of mean size during geological time, referred as the Cope's rule or Cope-Depéret rule, has been primarily showed on horses lineage but the observation has been extended to other mammals and marine organisms as well. It has been suggested that this increase of size is only a passive phenomenon because there is a lower functional limit in size (A limit under which the physiology of a given organisms cannot function), whereas the "upper" limit is less problematic.

Therefore, after each crisis where the smaller taxa survive (because less specialized and more likely to survive), the "average" size in a group increases but the median and the mode of the size remain low. This is what I observe on the Figure 3 where the darker colour indicates that most of the coccolith remained in the smaller part of the size spectrum, at least until 50 Ka where the diversification within *Gephyrocapsa* seems to be directed towards larger sizes (Except for *G. parvula* and *G. ericsonii*). Heim et al. (2015) (Heim, N.A., Knope, M.L., Schaal, E.K., Wang, S.C., Payne, J.L., 2015. Cope's rule in the evolution of marine animals. *Science*. 347, 867–870. doi:10.1126/science.1260065) tested the cope rule on the fossil record of marine organisms to answer whether or not the increase of size was a passive trend or not and they showed that the observe pattern could not be explained by a neutral drift. I think the authors could discuss the Cope's rule in their paper since it fits to the topic.

As a final remark, I have noted that the author are using data from Okada and Matsuoka (1990) and that they obtained the data from Prof. Matsuoka, however the data are not publicly available. These data should be made available (Through Pangaea for example: <https://www.pangaea.de/>) prior to the publication of the paper of El Bendif et al., and ideally the author should provide the result of their re-analysis as supplementary material as well.

Therefore I recommend the article to be publish but I feel necessary that the authors tone down certain claims in their discussion/conclusion and remain close to their results.

Detailed comments:

L.23 Maybe add "During the last 0.5 Ma".

L32-34. Could you please rephrase to accentuate the opposition in speciation process between land and marine ecosystems?

L53-54. I would remove the "which underpins their value in determining the age of sediments", this is not, in my opinion, relevant for the topic of the paper (although this is true).

L62. I would remove "macroevolutionary"

L67. Please provide the number of strains here instead of saying "diverse set of strains".

L69-71. I would remove this last sentence and finish the intro on "... extant patterns of genetic variation"

L104. Please explain why "nearly" and not all strains of the same morphospecies clustered together.

L120. Could you provide a number instead of "considerable proportion", even if you provide this number shortly after (11.6%).

L127. Could you remove the "long" before extinct? 0.5 Ma is not really long for most geologists.

L138. Could you provide a reference and a number for the estimation of population size?

L149. Could you please clarify what you mean with "speciation bottleneck"? I understand what it means but it could be clearer.

L151. Please name the ancestral species.

L186-187. Could you please explain in what consist these data? Morphometric data in itself is a bit vague... On the figure 3a I see only size measurements, and if so, then the author should mention only "size measurements" since morphometry aims at quantifying the shape. I also re iterate here that these data should be made available by the original authors

L197-198. I would skip the sentence on the disappearance event being used as a stratigraphic marker. Even if this is true, this disrupt the argumentation flow of the author.

L212-214. I do not understand the argument of the author.

L220-223. The author are suddenly talking about specific details about Coccolithophores morphology that can leave the non-expert reader (such as me) perplex. Could you either explain these characters or even provide a plate in the supplementary figures to highlight the morphological features you are talking about? Especially illustrate what a bridge is.

L252-258. This whole paragraphs needs more explanations. The authors are referring to similar results produced on coccolithofores during the Miocene, as well as similar observations made on other fossil plankton groups. Please explain what observations have been made and why they fit with your own observations.

L288-290. I am not convinced by this argument. Is there any evidence that oceanic front are effective barrier against gene flow. If yes, please cite it.

L291-300. I feel that this paragraph is somehow a bit void and that the authors are wandering too far away from their results. I think that the fact that biodiversity reacts to global climate change is widely accepted and that this part of the discussion is not necessary.

L303-306. I think that the statement starting the concluding remark is a bit bold. The authors, in my opinion, nicely showed the link between genomic and fossil data but I do not think that there is demonstration of the link between micro and macro evolutionary process. In short, the author may have showed the pattern, but not the process yet. I would remove this sentence and start directly by the second one, which is closer to the result of the study.

L312-316. Similarly, I do not think that this is a groundbreaking result to say that speciation in open ocean happens in sympatry and not in allopatry. I would either remove this sentence or tone it down.

L317-318. I think that it would be here safer to talk about pattern and not process. In general, I think that the author should insist more on the fact that there are probably radiation and extinction events provoking a high turnover in diversity through these pulses. It is in my opinion, more interesting.

L323-332. I don't have the feeling that the series of question that the authors are asking really adds value to the manuscript, it could stop at the line 323 and I would be satisfied by it.

L344-349. Please explain in what consist these data, and make sure they are available. Could the author explain what they mean when they say "... reanalyzed and related to modern taxonomy concepts of the species represented"? This sounds vague.

Figure 2C. Could you please write the names of the species next to the cartons at the bottom of the figure?

Figure 2D. Please show the cumulative abundance by stacking all the groups, it will be easier to read.

Figure 2E. If you provide the stacked relative abundances on the figure 2D, then I think that you can show the absolute abundance of all species together (single curve), it will be easier to read the patterns then.

Figure S3. Please plot the data point as well together with the smoothed curves.

20-July-2019

Reviewers' comments:

Reviewer #1 (Remarks to the Author):

This is a very well-written manuscript that is the first to combine the analysis of fossil coccolith data with genomic analysis of extant species to advance the understanding of plankton species radiation. It is novel and I pretty much appreciated the effort of the authors introducing such a detailed data-set including morphologic analysis, molecular phylogenetic work, comparison of the phylogenetic evolution to size variations of Gephyrocapsa coccoliths as well as a demographic modelling of Gephyrocapsa speciation history that provide strong evidence for its conclusions.

I see no problems in the work, which I like very much, and I believe that the integration of population and evolutionary genetics with diverse fossil records sheds novel light on the speciation mode in the recent evolution of a key phytoplankton group. I can follow essentially all the conclusions drawn from this study well, so that I recommend the manuscript for publication with minor revision only.

I have noted only minor suggestions and typos:

Title - Since the study relies exclusively on the coccolith genus Gephyrocapsa, it should, in my view, also find expression in the title of the paper.

Reply: "Gephyrocapsa" was added to the title

Line 51ff - The sentence "Coccolith sediments represent a rich source of nannofossils ..." reads a bit strange and should be rephrased. Furthermore, the authors should be aware that although many occurrence and extinction dates may appear to be globally synchronous, there are quite a number of examples where such an event is recorded in different stratigraphic positions in different oceanic areas (see Raffi et al., 2006 QSR).

Reply: The sentence on line 51 has been rephrased. The caveat regarding global synchronicity has been added.

Line 105ff – I wonder how the authors assess the morphologic study of Gephyrocapsa by Bollmann (1997) and the distinction of morphological groups within the genus characteristic of different

environments in the context of their findings!

Reply: Bollmann (1997) undertook a detailed survey of Gephyrocapsa morphological variation based on core-top sediment samples and based on this suggested a rather different taxonomy to that conventionally used, in particular his results suggested that G. oceanica was a complex of several species. His results have not been supported by subsequent workers. Instead, a range of work suggest that the conventional morphotaxonomy is robust, and this is supported by our results. It is likely that G. oceanica does show significant ecophenotypic variation but that genetically it is a single species.

Line 138 – It is not entirely true that “G. huxleyi is the most common coccolithophore ... for at least 70 ka”! Actually it is of global dominance since the beginning of the Holocene, but it is as abundant as other Gephyrocapsa species since 70 ka, as you can also see in Figure 2d.

Reply: We rephrased that sentence to read: “In particular, G. huxleyi, which has been globally abundant since the beginning of the Holocene and became as abundant as other Gephyrocapsa species 70ka [ref15], is now the most common coccolithophore in modern oceans (with a population size of the order of 10^{22} cells [ref41])...”

Line 185ff – The species radiation in Gephyrocapsa fossil record is reduced to variations in coccolith size. I can understand that it is necessary to break down such a development on individual characters, and I do not want to criticize it, but are there similar variations in other parameters? The species are not only defined by the size but also, according to Bollmann (1997), by the bridge angle.

Reply: In fact, we analysed three independent parameters: coccolith length, bridge angle and central opening width, which allowed us to document the evolution of coccolith morphology in detail. This is now clarified in the 1st sentence of the “...fossil record” section: “...we analysed the evolution of Gephyrocapsa coccolith morphology over the Pleistocene (1.8 Ma; Figure 3a) measuring coccolith size, bridge angle and central opening (Figure 3b; Table S5)...”

Line 260ff - I am not fully convinced that sea-surface temperature variations (alone) are likely to drive the species radiations and extinctions. I see a connection to the climate evolution but not necessary a causal relation (and doubt that this is possible to infer from the Δ SST).

Reply: We do not state that there is a direct causal relation between climate and radiations/extinctions of species. In fact, the paper clearly states that “MO events are too extended to follow the periodic Milankovitch glacial cycles of the Pleistocene”. The section the reviewer refers to, merely discusses possible connections between MO events and climate. DeltaSST is one of the most relevant proxies for climatic effects on marine plankton, hence it is useful to show it in fig2 and discuss possible association of MO events with climate.

Line 306 – I think it is not mentioned before how cell size and coccolith size are related? What results in the statement that changes in cell size are observed?

Reply: The correlation of coccolith and cell size has been reported in the literature (e.g. Hagino, K. & Young, J. R. in Marine protists: diversity and dynamics (eds Susumu Ohtsuka et al.) 311-330 (Springer Japan, Tokyo, 2015). We edited that sentence: “pulses of cell size change” were replaced with “pulses of coccolith (and likely cell) size change”. The references were also added.

Line 325 – In am a bit puzzled that the paper ends with a lot of questions for future research.

Reply: The questions at the end of the paper were removed

Line 344 – Its Stratophenic data ...

Reply: Unclear what the reviewer means here. The correct spelling is “stratophenetic”

Table 1 – its not G. margerelii but G. margereli (although both spellings are also used in Nannotax)

Reply: Corrected

Supplementary Figure S2 – I would recommend adding the names of the four Gephyrocapsa species here.

Reply: Species names added

Lines 677, 685 – Gephyrocapsa spp (spp not in italics)

Reply: Corrected

Reviewer #2 (Remarks to the Author):

The manuscript by Bendif et al. presents a study on the diversification patterns of haptophytes of the genus Gephyrocapsa. Despite the ecological importance of these species, this study is particularly interesting because it reports a comprehensive analysis of genetic and paleontological data. Together, these two lines of evidence picture an scenario of repeated speciation/extinction cycles in the last 2 million years. The integrative nature of the methodology and the conclusions of this study are very relevant for the community of evolutionary biologist.

I really enjoyed reading this paper. It is very well written. The results are presented in a logical way and the methodology seems (mostly) robust. I only have two major concerns regarding the methodology: the molecular clock analyses and the rooting of Gephyrocapsa phylogeny with Tysochrysis.

Molecular clock. The methods of this section are very poorly described and key important settings are not mentioned. Which type of relaxed molecular clock was used, autocorrelated or uncorrelated? How was the calibration specified? Calibrations should always incorporate its inherent uncertainty (e.g. with a uniform prior with min and max ages). What was the assumed prior for the overall rate parameter (rgene_gamma)? This parameter has been shown to have a large effect on posterior time estimates (Dos Reis et al. 2014 Syst Bio 63). Were timetrees inferred with full or approximated likelihood? Please, specify all relevant parameters. In addition, because the divergence times are one of the key results on which the paper is built, it would be ideal if the authors could assess the robustness of their results using some sensitivity analyses, such as comparing the effect of using different clock models, different calibrations (if available), etc.

Reply: We rewrote the methods section devoted to molecular clock. Now it shows the details of the parameters used in the analyses. In particular, we used uniform prior with min=290ka and

*max=350ka as a calibration for node 2; no other reliable calibration points are available. All analyses were done with full likelihood. Furthermore, we re-ran the analysis with different settings (correlated and uncorrelated clock and a range of *rgene_gamma* values, see Table S6), which confirmed that the dating of the nodes is robust to variation in parameter values.*

Rooting. Supplementary Fig. S1 suggest a very long branch separating *Tisochrysis* from *Gephyrocapsa*, which led me wonder if there could not be a “random rooting” effect. When the outgroup is very distant, its sequence likely contains insufficient information on the polarity of characters within the ingroup and can led to the long branch of the outgroup being connected in any branch in the ingroup with low reliability, “randomly” (Wheeler 1990 *Cladistics* 6; Hirase et al. 2016 *Genome Biol Evol* 8). If this were the case, changing the root of the *Gephyrocapsa* phylogeny would have a huge impact on molecular clock. This could be alleviated by using a closer outgroup (if available) or a broader sampling of outgroups.

*Reply: We added the second outgroup - *Isochrysis galbana* strain CCMP1323. Both outgroups consistently place the root of the tree on the branch connecting *G. oceanica* with all other *Gephyrocapsa* species used in the study.*

In relation to this rooting problem, the presence of undetected paralogs in *Tisochrysis* might exacerbate the already long branch, leading to random rooting artifacts. The authors identify homologous sequences in *Tisochrysis* as best blast hits with the other species. However, this strategy alone is not enough to ensure that the sequences are orthologous and the authors do not report any step towards identifying and removing possible paralogs. It is known that the inclusion of a few paralogs can negatively affect phylogenomic inference and molecular clock analyses (e.g. Shen et al. 2017 *Nat Ecol Evol*; Siu-Ting et al. 2019 *Syst Bio*). The authors could exclude paralogs using a broader taxon sampling in the and using phylogenetic inference of locus trees, for example. Also, the use of more complex mixture models that often fit the data better (e.g. CATGTR or C60 family) could also help assess the reliability of the position of the root.

*Reply: We repeated this analysis after replacing best-blast-hit approach with Orthofinder, which takes the possibility of paralogy into account. The results of this re-analysis are consistent with the previously used best-blast-hit approach: in both cases the root is placed on the branch leading to *G. oceanica*. Using CATGTR also supported this placement of the root. This is now explained in the corresponding methods section.*

Other remarks:

The correspondence between extant *Gephyrocapsa* species and large/medium/small species from the cores (e.g. in Figure 2) is unclear, at least for me as a non-specialist in the system.

*Reply: We added the following clarification to fig2d legend: “Large *Gephyrocapsa* include *G. oceanica*; Medium, *G. margereli* and *G. muelleriae*; Small, *G. aperta* and *G. ericsonii*.”*

For example, in Table 1, column “Remarks”, this is particularly surprising: what does it mean that *G. lumina* is “sometimes referred to as large *Gephyrocapsa* and *G. oceanica*” etc.? Is it because determining morphospecies in fossils is difficult or because the taxonomy has been changing and so species names are not comparable across studies? It would be great if the authors could clarify this for non-specialists.

*Reply: The correspondence between extant species and species described from sediment data is not always easy to establish and it depends on the species. In some cases, such as *G. huxleyi*, species*

*identity is clear both in extant and fossil data due to very distinctive morphology, while in other cases it is not always possible to establish species identity with certainty. The use of different names by different authors to describe fossil data does not make the situation simpler. To clarify this in the paper we added a note under that table: “**Species names in the fossil record are not always consistent across the studies”. Furthermore, to better illustrate the morphology of different species we added suppl Figure S1, which made this table partly redundant so we moved it to supplementary (now Suppl table S7).*

Similarly, several terms specific to coccolithophorids are mentioned in the paper (coccolith length and presence of bridge; lines 241-251). Could you please briefly explain what they are and what their relevance is?

Reply: We added a panel to fig3 (panel 3b) to show the characters measured

Data and code availability. The datasets (including raw morphometric data) and code (including the pipeline for testing for topological incongruences, line 435) should ideally be made available as supplementary material or deposited in recognized repositories (e.g. dryad, figshare) to ensure reproducibility and avoid that this data might be lost for the scientific community.

Reply: The sequenced data was uploaded to NCBI; morphometric data was uploaded to Pangaea database. The scripts used in the paper are available from github page of the 2nd author: <https://github.com/brunonevado/> With regard to testing for topological incongruence, we did not use any scripts that could be re-used by others.

Sampling locations. The localities where sediments and living strains were obtained are geographically disparate (Figure 1). Could this have affected the definition of morphospecies in either the sediments or the sequenced strains, or hinder some important local effect?

Reply: As the species have global distribution and show global synchronicity, we consider it unlikely that different locations for sediments and sequenced strains would make a difference. It is logistically impossible to collect samples only at the locations where sediment cores were obtained.

Methods: DNA extraction. How many cells were used for sequencing? Or can this be approximated somehow, e.g. dry weight?

Reply: DNA extraction was done from ~0.05g of cell pellet from clonal cultures (~100 million cells), but only a small fraction of the resulting DNA was used for sequencing, so it is not possible to tell how many cells were used “for sequencing”. Given the cultures were clonal, the actual number of cells used does not affect the results of our analyses in any way.

Methods: read mapping. It would be great to have a supplementary table with details on the genome data, including mapping statistics per genome. Also, 40% read mapping might seem quite low (unacceptable in most animals and plant systems). I assume this is due to the known variability of *G. huxleyi* pan-genome (Read et al. 2013 Nature 499), is it correct? Again, this might not be obvious for the non-specialist and merits a brief explanation.

Reply: This information was added to suppl table S1 and clarified in the methods: “Despite low sequence divergence (<3% total sites) between the strains analysed, the proportion of reads mapped

to reference was relatively low (40-69%; Suppl. Table S1) likely due to the known variability of G. huxleyi pan-genome with a large proportion of the genome missing in many strains."

Methods: coalescent species tree. ASTRAL analysis are based on 10kb windows. These are quite large regions and ASTRAL assumes no recombination within each of these 10kb windows, which seems unlikely. Could this have affected the ASTRAL tree?

Reply: we re-run the ASTRAL analysis with shorter windows, which yielded the same result (Figure S4). This is now clarified in the methods section describing ASTRAL analysis: "The same ASTRAL analysis was repeated with shorter (5kb long) windows, which yielded the same species tree as the analysis with 10kb long windows (Figure S4)."

Figure 1. Add number and unit to the unit scale (to avoid being confused with the lines delimiting species).

Reply: number and unit added to fig1b

Figure 2 caption. is deltaSST a global measurement or only of site 1082?

Reply: deltaSST is based on global measurements – now clarified in fig2 legend

Supplementary Fig. S4. Replace "um" by the right symbol nu.

Reply: We replaced "um" with μm .

Supplementary Fig. S5. Rainbow color scheme is misleading for representing continuous measurements such as length and probability. A simple grayscale would do it. Also include species names (not only vouchers) and specify coding for bridge (I assume 1=presence and 0=absence).

Reply: Figure S5 (now figure S9) was modified accordingly

Supplementary Table S4. The results from ABBA/BABA tests could be better represented graphically than in a table, e.g. using violin plots (see Irisarri et al. 2018 Nat Commun 9, Fig 3 <https://www.nature.com/articles/s41467-018-05479-9>).

Reply: We added violin plot in suppl figure S4 to make the results of ABBA/BABA test more visual

Lines 61-62: this sounds like Cenozoic Reticulofenestra would be ancestors of Gephyrocapsa species, which makes no sense given that Reticulofenestra are also extant species. Could this be rephrased?

Reply: "ancestors" replaced with "relatives"

L87-88: How were morphospecies delimited? Was it based on a phylogenetic analysis of morphological data? If so, please present the tree in the supplement. Also, the underlying morphological data should be made available in a recognized repository. Or is this data in Supplementary Table S5? This table is also not mentioned in the text.

Reply: To clarify this point we added suppl figure S1 and references to previous work: “Based on morphology and previous studies (Bendif et al 2015, 2016), these 10 Gephyrocapsa strains were classified as belonging to five different species (Figures 1b and S1), covering almost the full range of extant morphological diversity in this genus.” Table S5 is now mentioned in the text. As mentioned above, morphometric data were submitted to Pangaea database.

L124: incongruence patterns can also derive from other phenomena, including introgression, as it is revealed by the coalescence analyses by the authors themselves (Fig. 2c).

Reply: This sentence was rephrased to read “incongruence can be caused by phenomena such as incomplete lineage sorting”

L235 (Fig 3 caption): aren't > are not

Reply: Corrected

L268-270: the coincident patterns between the morphospecies successions and planetary cooling are not very obvious for me from Fig 2d-f.

Reply: We do not claim that there is a strong association with climate; this is only a discussion point. The sentence reviewer points to has been removed

L296: hominids should not be capitalized.

Reply: Corrected

L306: I think a softer phrasing such as “Our study suggests” or similar would be better here. This conclusion depends on the credibility of the molecular clock analysis.

Reply: Corrected: “revealed” changed to “indicated”

L325 Regarding: “Why does diversification lead to increased range of sizes?” Could it be possible that size increase is only one of the obvious patterns that we recognized as speciation?

Reply: The observed diversification was not only in the range of sizes, but also in bridge angle and the size of central opening, as now explained in the paper. With regard to the increased range of sizes, we added a paragraph discussing this point in the context of Cope’s rule: “Within each MO-cycle we see the pattern of gradual size increase, followed by abrupt decrease. It has been suggested that such increase in size, often referred to as Cope’s rule, is only a passive phenomenon caused by a lower physiological limit in size. Therefore, after each extinction event, when only the smaller less specialized taxa survive, the maximal size in a group gradually drifts upwards, while the median and the mode of the size remain low, as can be see on Figure 3a, where the darker colour indicates that most of the coccoliths remain in the smaller part of the size spectrum, while the upper limit increases. However, Heim et al, who tested the Cope’s rule on the fossil record of marine organisms, demonstrated that the increase in size could not be explained by a neutral drift and thus is possibly driven by selection for larger size.”

L328: Regarding: “Are the cycles/extinctions driven by external factors such as climate/cooling as suggested here, or are the cycles internally self-limiting perhaps as a result of the increase in size and

pressure on some limiting resource" ... or maybe both?

Reply: We agree that both are possible, so we replaced "...or..." with "...and/or..."

L355: either ... and

Reply: Corrected

L374, 401, 692 and others: "maximum likelihood", "parsimony" or "incomplete lineage sorting" should not be capitalized. Also, introduce abbreviations on first time of appearance rather than in the methods section.

Reply: Corrected

L380-381: Could you clarify what you mean with "after excluding the alignments with less than 80% of sites without missing data"? Were sites with >20% excluded or instead entire 10kb alignments when they had <80% missing data? Were gaps considered missing data?

Reply: This is now clarified in the methods: "...after excluding the alignments positions with >20% of gaps or missing data"

L394: blastp

Reply: Corrected

References. Format should be homogenized.

Reply: Corrected

Reviewer #3 (Remarks to the Author):

The article entitled "Repeated species radiations in the recent evolution of a key marine phytoplankton lineage" presents a study of the evolution modus of phytoplankton. The authors have generated genome sequencing of 10 strains of the *Gephyrocapsa* genus and used these data to reconstruct the evolutionary history of the constitutive species. The author compare their results to the fossil record of the coccolithophores and highlight a pattern of repeated species radiation that they designate as the "Matsuoka-Okada" (MO) cycles that seem to have a cyclicity of about 0.5 Ma. I have appreciated to read the article. The author have generated a comprehensive genomic dataset, compared their result to the patterns of the fossil record, and showed many points of convergence. The writing is clear, the illustrations of high quality and the author show clearly new patterns of Coccolithofores evolution. I have only critics on the last third of the article where the author are discussing what drives repeated plankton species radiations and extinction and my impression is that this section of the article is weaker. This is because the authors try to link paleoclimatic changes to the MO cycles but this does not constitute a solid argumentation in my opinion, and I think that solving these questions would require an independent study with statistical tool and comparison with parameters other than the SST.

Reply: We agree that solving these questions would require a separate study. That is why in our paper we only discuss our results in the context of changing climate through time and we do not

claim that the MO-cycles are directly driven by climatic changes. We hope this is clear in the revised version of the paper.

In addition, the author rule-out pure allopatric speciation as a mode of speciation in Gephyrocapsa but this does not sound to me as a groundbreaking discovery because, as the author mentioned it in their introduction, the lack of geographic barrier in the ocean alluded speciation to sympatric mode.

Reply: Despite perceived lack of barriers for gene flow in the sea, allopatric speciation may still occur and might actually be very common as geographic, geologic and oceanographic (e.g. sea fronts or ocean depth) barriers do exist and can separate plankton populations. Thus, it is essential to test the intuitive expectation about speciation mode in the ocean with the actual results, which is done in our study.

Similarly, they authors ask in their conclusion why diversification lead to an increase of size. This is, in my opinion, not a novel question and it has actually produced a prolific literature already. This phenomenon of increase of mean size during geological time, referred as the Cope's rule or Cope-Depéret rule, has been primarily showed on horses lineage but the observation has been extended to other mammals and marine organisms as well. It has been suggested that this increase of size is only a passive phenomenon because there is a lower functional limit in size (A limit under which the physiology of a given organisms cannot function), whereas the "upper" limit is less problematic. Therefore, after each crisis where the smaller taxa survive (because less specialized and more likely to survive), the "average" size in a group increases but the median and the mode of the size remain low. This is what I observe on the Figure 3 where the darker colour indicates that most of the coccolith remained in the smaller part of the size spectrum, at least until 50 Ka where the diversification within Gephyrocapsa seems to be directed towards larger sizes (Except for *G. parvula* and *G. ericsonii*). Heim et al. (2015) (Heim, N.A., Knope, M.L., Schaal, E.K., Wang, S.C., Payne, J.L., 2015. Cope's rule in the evolution of marine animals. *Science*. 347, 867–870. doi:10.1126/science.1260065) tested the cope rule on the fossil record of marine organisms to answer whether or not the increase of size was a passive trend or not and they showed that the observe pattern could not be explained by a neutral drift. I think the authors could discuss the Cope's rule in their paper since it fits to the topic.

Reply: We now mention Cope's rule at the end of introduction: "The gradual size increase may be an instance of general Cope rule – the trend of organism size increase within a lineage with evolutionary time, which was first described for terrestrial animals[21], but has also been reported in marine realm[22]." Furthermore, we added a paragraph to the "Repeated species radiations..." section to discuss our results in the context of Cope's rule: "Within each MO-cycle we see the pattern of gradual size increase, followed by abrupt decrease. It has been suggested that such increase in size, often referred to as Cope's rule..."

As a final remark, I have noted that the author are using data from Okada and Matsuoka (1990) and that they obtained the data from Prof. Matsuoka, however the data are not publicly available. These data should be made available (Through Pangaea for example: <https://www.pangaea.de/>) prior to the publication of the paper of El Bendif et al., and ideally the author should provide the result of their re-analysis as supplementary material as well.

Reply: The data of Okada and Matsuoka (1990) have been uploaded to Pangaea database and will be publicly available shortly. We prefer not to include these data as supplementary in the study as this is not our data and it is more appropriate to make them public via Pangaea database.

Therefore I recommend the article to be published but I feel necessary that the authors tone down certain claims in their discussion/conclusion and remain close to their results.

Detailed comments:

L23 Maybe add “During the last 0.5 Ma”.

Reply: Corrected

L32-34. Could you please rephrase to accentuate the opposition in speciation process between land and marine ecosystems?

Reply: The first paragraph of introduction already outlines the difference between the speciation processes on land and in the sea: “Adaptation and speciation processes may work in rather different ways in relatively small subdivided populations of terrestrial organisms and globally ubiquitous populations of abundant marine plankton. In particular, it is unclear how new plankton species form in relatively homogenous habitats, such as in the open ocean[2], where no physical barriers to gene flow exist to promote allopatric speciation (i.e. the passive divergence of isolated populations) that is considered to be the most common speciation scenario in terrestrial organisms[5,6]”.

L53-54. I would remove the “which underpins their value in determining the age of sediments”, this is not, in my opinion, relevant for the topic of the paper (although this is true).

Reply: Corrected

L62. I would remove “macroevolutionary”

Reply: Corrected

L67. Please provide the number of strains here instead of saying “diverse set of strains”.

Reply: We added “10 strains” but kept “diverse set” because we feel it is important to stress the diversity of strains analysed

L69-71. I would remove this last sentence and finish the intro on “... extant patterns of genetic variation”

Reply: Corrected

L104. Please explain why “nearly” and not all strains of the same morphospecies clustered together.

Reply: This is already explained two sentences later in the same paragraph

L120. Could you provide a number instead of “considerable proportion”, even if you provide this number shortly after (11.6%).

Reply: We added “(11.6%)” right after “considerable proportion”

L127. Could you remove the “long” before extinct? 0.5 Ma is not really long for most geologists.

Reply: Corrected

L138. Could you provide a reference and a number for the estimation of population size?

*Reply: We have to note that any such estimates are speculative. We added a citation of Emiliani’s 1993 paper, which states that *E.huxleyi* is currently the most abundant plankton species, with population size of the order of 10^{22} .*

L149. Could you please clarify what you mean with “speciation bottleneck”? I understand what it means but it could be clearer.

Reply: Added. Now the end of the sentence reads: “...without speciation bottleneck (i.e. without a period of reduced population size due to origination of a species in a very small population)”.

L151. Please name the ancestral species.

*Reply: We cannot be more precise than just saying “ancestral *Gephyrocapsa* species” – now in the paper.*

L186-187. Could you please explain in what consist these data? Morphometric data in itself is a bit vague... On the figure 3a I see only size measurements, and if so, then the author should mention only “size measurements” since morphometry aims at quantifying the shape. I also re iterate here that these data should be made available by the original authors

*Reply: We added a panel to fig3 (fig3b) and modified the sentence in question to read: “Using morphometric data from Matsuoka & Okada30 we analysed the evolution of *Gephyrocapsa coccolith* morphology over the Pleistocene (1.8 Ma; Figure 3a) measuring coccolith size, bridge angle and central opening (Figure 3b; Table S5) in successive populations with a sampling resolution of about 40 ka.” The data were submitted to Pangaea database by prof Matsuoka.*

L197-198. I would skip the sentence on the disappearance event being used as a stratigraphic marker. Even if this is true, this disrupt the argumentation flow of the author.

Reply: We opted to keep this sentence as it stresses that the event is globally synchronous and illustrates the importance of this event for wider applications – significant stratigraphic datum for oil industry and ODP/IODP program etc.

L212-214. I do not understand the argument of the author.

Reply: We argue that if MO are caused by change in abundance of the same species, then their common ancestor is expected to be very old. In other words, let's say we have one big-shelled (B) and one small shelled species (S) co-existing in world oceans for the last ~2 million years. If their abundance fluctuates through time and peaks of B correspond to troughs of S, then we'll see a pattern similar to MO cycles – pulsing of shell size. If this model were correct, then the common ancestor of B and S should be older than 2MY because no speciation occurs in this model of B and S co-existence. The alternative model is that MO cycles correspond to speciation events. In that model the common ancestor should be young, with the age corresponding to the last MO – exactly what we see in the data.

L220-223. The author are suddenly talking about specific details about Coccolithophores morphology that can leave the non-expert reader (such as me) perplex. Could you either explain these characters or even provide a plate in the supplementary figures to highlight the morphological features you are talking about? Especially illustrate what a bridge is.

Reply: The details of coccolith morphology are now shown on figures 3b and S1.

L252-258. This whole paragraphs needs more explanations. The authors are referring to similar results produced on coccolithofores during the Miocene, as well as similar observations made on other fossil plankton groups. Please explain what observations have been made and why they fit with your own observations.

Reply: The paragraph was rewritten: "Further back in the fossil record, events similar to MO-cycles have punctuated the evolution of Gephyrocapsa ancestors over at least the past 15 Ma^{49,50}. Other plankton groups including the diatoms²³, dinoflagellates²⁴ and planktic foraminifera²⁵ all demonstrate patterns of macroevolutionary variation in size attributed to changing upper ocean structure over similar timescales. Finer scale evolutionary events in the Gephyrocapsa ancestry, with characteristics similar to MO-cycles, have also been identified, such as periods of dramatic size reduction⁴³. Furthermore, the history of the coccolithophore genus Calcidiscus displays the repeated development of coccoliths similar in appearance but at stratigraphically distant intervals⁴⁴. These observations indicate that pulsed events in plankton evolution may be widespread across the coccolithophores and our results indicate that they represent species radiations separated by abrupt extinctions."

L288-290. I am not convinced by this argument. Is there any evidence that oceanic front are effective barrier against gene flow. If yes, please cite it.

Reply: There is ample evidence to suggest that oceanic fronts act as a physical barrier to gene flow between species of benthic, littoral and demersal organisms: diatoms (Casteleyn et al 2010 PNAS 107: 12952–12957), foraminifera (de Vargas C, et al. 2002 Mar Micropaleontol 45:101–116; Darling KF, Wade CA 2008 Mar Micropaleontol 67:216–238; Aurahs et al 2009 Mol Ecol 18:1692–1706), fish (Galarza et al 2009 PNAS 106: 1473–1478) and molluscs (Sa' Pinto 2012 Plos1 7: e50330). We added some of these refs to the paper.

L291-300. I feel that this paragraph is somehow a bit void and that the authors are wandering too far away from their results. I think that the fact that biodiversity reacts to global climate change is widely accepted and that this part of the discussion is not necessary.

Reply: We shortened this paragraph. We feel it is important to link the patterns observed in Gephyrocapsa to that in other domains, thus we opt to keep a shorter version of this paragraph.

L303-306. I think that the statement starting the concluding remark is a bit bold. The authors, in my opinion, nicely showed the link between genomic and fossil data but I do not think that there is demonstration of the link between micro and macro evolutionary process. In short, the author may have showed the pattern, but not the process yet. I would remove this sentence and start directly by the second one, which is closer to the result of the study.

Reply: We agree that we only start to build a link between macro-patterns and micro-evolutionary processes, hence we dropped the second part of that sentence.

L312-316. Similarly, I do not think that this is a groundbreaking result to say that speciation in open ocean happens in sympatry and not in allopatry. I would either remove this sentence or tone it down.

Reply: We toned it down by replacing “shed the first light” with “reveal”.

L317-318. I think that it would be here safer to talk about pattern and not process. In general, I think that the author should insist more on the fact that there are probably radiation and extinction events provoking a high turnover in diversity through these pulses. It is in my opinion, more interesting.

Reply: We toned down the 1st sentence in the last paragraph by replacing “provide a link” with “help to build a link”. The note about “high turnover in diversity” suggested by the reviewer was added to the sentence in the 1st paragraph of conclusions: “Similar patterns of cycles in size appear to be common in many plankton groups[48] and may also represent repeated pulses of species radiations and extinctions that lead to high turnover in diversity.”

Throughout the paper we use “process” in the micro-evolutionary context, where processes shaping genetic diversity and driving microevolution are reasonably well understood thanks to population genetic work in terrestrial organisms. On the other hand, in macroevolutionary context we always use “pattern(s)”, as pointed by the reviewer.

L323-332. I don't have the feeling that the series of question that the authors are asking really adds value to the manuscript, it could stop at the line 323 and I would be satisfied by it.

Reply: Questions at the end of conclusions were removed.

L344-349. Please explain in what consist these data, and make sure they are available. Could the author explain what they mean when they say “... reanalyzed and related to modern taxonomy concepts of the species represented”? This sounds vague.

Reply: We added figure 3b to show the traits measured. To make the morphometric data publicly available we asked prof Matsuoka to upload the data to Pangaea database, which was done. The re-analysis of these data is described in the results. “Related to modern taxonomic concepts” refers to the re-interpretation of the data in terms of the species names used in the modern literature; it was

changed to "...data was reanalysed and re-interpreted in the context of modern taxonomic concepts of the species represented".

Figure 2C. Could you please write the names of the species next to the cartons at the bottom of the figure?

Reply: Species names were added to figure 2C

Figure 2D. Please show the cumulative abundance by stacking all the groups, it will be easier to read.

Reply: The figure 2D was modified accordingly

Figure 2E. If you provide the stacked relative abundances on the figure 2D, then I think that you can show the absolute abundance of all species together (single curve), it will be easier to read the patterns then.

Reply: The figure 2e shows the absolute abundance of all species together

Figure S3. Please plot the data point as well together with the smoothed curves.

Reply: The figure S3 (now S7) was modified accordingly

Reviewers' Comments:

Reviewer #2:

Remarks to the Author:

The revised version of the manuscript addressed all my concerns and I have further comments, besides the need of adding the DOI of the data uploaded to the Pangaea database. Therefore, I recommend this article to be published.

Reviewer #3:

Remarks to the Author:

The article by Bendif et al. has been significantly improved after the revision. The authors have acceded to nearly all my requests and when they did not, they provided excellent justifications. I have seen that the authors did the same for the comments of the two other reviewers. After a detailed reading of the manuscript, I have no further comments to add. I am very pleased to recommend the article "Repeated species radiations in the recent evolution of a key marine phytoplankton lineage" for publication and congratulate the authors for the excellent manuscript they have produced.